biomechanics/computer modelling and simulation/bioengineering

haemodynamics, atherosclerosis, mechanobiology

**Author for correspondence:**
P. Sáez
e-mail: pablo.saez@upc.edu

# Mechanotransmission of haemodynamic forces by the endothelial glycocalyx in a full-scale arterial model

## P. Sáez[1,2], D. Gallo[3] and U. Morbiducci[3]

[1]Laboratori de Càlcul Numèric (LàCaN), Universitat Politècnica de Catalunya, Barcelona, Spain
[2]Barcelona Graduate School of Mathematics (BGSMath), Barcelona, Spain
[3]PoliToBIO Med Lab, Department of Mechanical and Aerospace Engineering, Politecnico di Torino, Torino, Italy

PS, 0000-0002-9253-0417; DG, 0000-0002-7409-7111; UM, 0000-0002-9854-1619

The glycocalyx has been identified as a key mechano-sensor of the shear forces exerted by streaming blood onto the vascular endothelial lining. Although the biochemical reaction to the blood flow has been extensively studied, the mechanism of transmission of the haemodynamic shear forces to the endothelial transmembrane anchoring structures and, consequently, to the subcellular elements in the cytoskeleton, is still not fully understood. Here we apply a multiscale approach to elucidate how haemodynamic shear forces are transmitted to the transmembrane anchors of endothelial cells. Wall shear stress time histories, as obtained from image-based computational haemodynamics models of a carotid bifurcation, are used as a load and a continuum model is applied to obtain the mechanical response of the glycocalyx all along the cardiac cycle. The main findings of this *in silico* study are that: (1) the forces transmitted to the transmembrane anchors are in the range of 1–10 pN, which is in the order of magnitude reported for the different conformational states of transmembrane mechanotranductors; (2) locally, the forces transmitted to the anchors of the glycocalyx structure can be markedly different from the near-wall haemodynamic shear forces both in amplitude and frequency content. The findings of this *in silico* approach warrant future studies focusing on the actual forces transmitted to the transmembrane mechanotransductors, which might outperform haemodynamic descriptors of disturbed shear as localizing factors of vascular disease.

# 1. Introduction

There is clear evidence that endothelial cells (ECs) respond to shear forces with a variety of mechanotransduction processes that lead to biophysical, biochemical and gene regulatory changes [1–4], with important implications in terms of cardiovascular pathologies [5–8]. In particular, Dai *et al.* [9] reported differences in ECs morphology, cytoskeletal actin organization and proinflammatory gene expression when exposing ECs to low and oscillatory versus relatively uniform shear force waveforms taken from a carotid artery model. Such mechanotransduction processes require a complex and well-orchestrated two-way communication between ECs and their environment, but are still poorly understood [4]. At the intracellular side, the actomyosin cytoskeleton shapes the cell [10], while at the extracellular side, a large number of proteins connect the cytoskeleton with the outside of the cell, serving as mechanosensors and mechanotransductors. In particular, a variety of actin-binding proteins are present at the subcellular level, linking the transmembrane proteins with the actomyosin cytoskeleton.

A primary role in the ECs mechanosensing of shear forces is played by the endothelial glycocalyx (GCX) [3,11,12], whose biological role is tightly related with atherosclerosis [13]. The GCX, a conglomerate of proteins and macromolecules lining the apical side of the ECs membrane, exhibits a very distinctive response to external stimuli, i.e. highly dynamic fluid forces, which are not present in any other cell type. How these mechanical forces are transmitted from the streaming blood to the anchoring elements at the GCX is still poorly understood [11,14,15]. Structurally, the GCX is a dense brush-like layer (see [4] for a review) containing several proteoglycans (PGs), and the attached glycosaminoglycans (GAGs). A major PG class are the glypicans that are not transmembrane but anchored into the membrane. They have been shown to initiate important signalling (e.g. nitric oxide) [16]. Core proteins are endowed with GAGs, hyaluronic acid (HA), heparan sulfate (HS) and chondroitin sulfate (CS), the latter deemed to have a major role in mechanosensation mechanisms [17,18]. HS contains two major protein core families, membrane bound glypicans [19,20] and transmembrane syndecans (syndecans-1, -2 and -4) [21,22]. Since syndecans (and in general the GCX anchoring proteins) emanate from the underlying actin cortical cytoskeleton, the GCX is responsible for the initiation of signaling [21], inducing the re-organization of the cytoskeleton [23] and the subsequent EC morphological changes, that could ultimately promote atheroma plaque formation [19,24]. In normal conditions, the GCX senses the haemodynamic shear through its extended surface and transmits it to the cell through the core proteins decorating the surface of the cell. Based on theoretical considerations [25], blood velocity is greatly attenuated within the GCX layer and hence it is vanishingly low at the apical EC membrane. Accordingly, the blood shear force acting directly on the EC apical membrane is negligible [4,25]: as a consequence, all the shear force acts as a drag on the GCX luminal edge, resulting in a bending moment on the GCX structures that is converted into mechanical stress at the anchoring proteins. More detailed models of the GCX have been proposed based on molecular dynamics [26,27]. Other coarse-grained models using numerical simulations, including Monte Carlo [28], brush-like [29,30] and Dissipative Particle Dynamics methods [31], have been proposed to depict the mechanical response of the GCX. However, the application of most detailed models to study the dynamic GCX response on the time and spatial scale here considered is computationally unfeasible.

In the transmission of these mechanical forces and deformations to the anchoring elements at the membrane of ECs, there is still a knowledge gap in our understanding of the involved force levels and, mostly, the dynamic behaviour of the fluid forces sensed by the GCX and transmitted at the EC membrane level.

In this study, a multiscale approach is applied to explore how haemodynamic shear forces are transmitted to the transmembrane anchors of ECs via the mechanical response of the GCX layer. For such a purpose, haemodynamic shear force time histories were first obtained through image-based computational haemodynamic modelling at the luminal surface of a carotid bifurcation, a vascular district prone to atherosclerotic lesion development. In this way, a large representative variety of realistic haemodynamic shear force time histories is obtained, covering the range from atheroprotective to atheroprone phenotypes. Secondly, the HS (i.e. one of the main mechanosensors in the GCX structure) are simulated with a continuum mechanical model based on the Timoshenko beam theory to obtain the simplified mechanical behaviour of the GCX layer under the action of the haemodynamic shear forces at the carotid bifurcation. We extended previous continuum approaches [25] with the addition of the inertial terms to model the mechanical response of the GCX along the whole cardiac cycle ($\approx 1$ s). Finally, the dynamic forces transduced to the anchoring elements of the HS on the EC membrane are evaluated and compared to the haemodynamic shear forces.

# 2. Methods

To analyse the transmission mechanism of blood shear forces from the floating tip to the GCX anchoring proteins at the EC membrane, a two-step, multi-scale approach is applied: (1) image-based computational haemodynamics is used to obtain the distribution of wall shear stress (WSS, i.e. shear force per unit area) at the luminal surface of a realistic model of human carotid bifurcation; (2) WSS time histories as obtained from haemodynamic simulation were prescribed as input loads to a simplified GCX model, based on the Timoshenko beam theory. This two-step procedure allowed the calculation of the reaction forces at the anchoring core proteins of the GCX that are linked to the cytoskeleton through syndecans. Lastly, the stimulating time histories and the GCX reaction forces were analysed both in the time and frequency domain, to gain insights on the transmission pathway of the haemodynamic shear forces to the EC membrane along the cardiac cycle.

## 2.1. Image-based haemodynamic simulation of the healthy human carotid bifurcation

Computational fluid dynamics represents an effective and affordable technique to reproduce the realistic flow patterns in three-dimensional reconstructed arteries, in particular, the shear stress distribution at the luminal surface, whose *in vivo* evaluation suffers from limitations [32]. Here, we used computational haemodynamics to analyse the distinct features of blood flow at the carotid bifurcation. Technically, an ostensibly healthy carotid bifurcation geometry was reconstructed from black blood MRI images [33,34]. Assuming blood as an incompressible, homogeneous and Newtonian fluid (with density $\rho$ equal to $1060 \, \mathrm{kg \, m^{-3}}$, and dynamic viscosity equal to 3.5 cP) the governing equations of fluid motion under unsteady flow conditions are

$$\left. \begin{array}{c} (\rho(\partial_t \mathbf{v} + \mathbf{v} \cdot \nabla \mathbf{v}) = -\nabla p - [\nabla \cdot \boldsymbol{\tau}] \\ \nabla \cdot \mathbf{v} = 0, \end{array} \right\} \tag{2.1}$$

and

where the velocity $\mathbf{v}$, the pressure $p$ and the deviatoric stress tensor $\boldsymbol{\tau}$ were discretized and numerically solved by applying the finite volume method. To do that, the CFD code Fluent (ANSYS Inc., Canonsburg, PA) was used on a fluid domain discretized by using tetrahedrons (mesh-grid cardinality equal to 1 400 000). Arterial walls are assumed to be rigid. The pulsatile flow rate waveform measured elsewhere [35] was applied at the common carotid artery (CCA) inlet section. The flow rate waveforms at the internal carotid artery (ICA) and external carotid artery (ECA) outlet sections were set to a fixed fraction of the CCA waveform (60% for ICA, 40% for ECA [34,35]). To ensure fully developed velocity profiles at the CCA inlet and to minimize the influence of outlet boundary conditions at the ICA and ECA, flow extensions were added to the inlet and outlet faces. Details on numerical settings have been extensively presented elsewhere [34,36].

## 2.2. Continuum model of the glycocalyx layer

An idealized model of the GCX structure is built up, based on experimental data available from the literature [25,37]. More refined models of the GCX have been proposed in the literature, for example, based on large-scale molecular dynamics simulations [26,27]. However, these simulations are limited in the time scale of investigation (in the order of dozens of nanoseconds), while here the considered time scale is in the order of 1 s, i.e. the duration of one cardiac cycle. The GCX, as mentioned above, is made of a plethora of GAGs, the most important ones, in terms of mechanotransmission, being syndecans and HS [4,17,18]. Here we focused on HS thin structures, as they are directly exposed to the fluid flow and transmit the friction forces related to blood flow to ECs through the transmembrane syndecans. Then, we computed the reaction forces in the syndecans which are anchored across the cell membrane.

In the past, a simplified continuum model was proposed [25], where the structure of the endothelial GCX was considered as a three-dimensional bush-like structure with a constant spacing of 20 nm in all directions and a diameter of 10–12 nm, anchored to the EC membrane [37,38]. Based on the observed hexagonal-like distribution (with a spacing in the order of 100 nm) of the anchoring locations [39–41], a model of the structural organization of the endothelial layer was suggested [37], and then used to ameliorate a previous continuum mechanics model of the endothelial GCX [25] by considering the inertial term. Although it describes a simple approach for the glycocalyx modelling, it is a reasonable model to explore fluid shear transmission to EC membrane all along the cardiac cycle. Here we adapt

the model to a single branch centred in the anchor, a level of idealization justified by the assumption that the same haemodynamic shear stress acts on all the elements of the hexagonal structure. The Timoshenko beam theory [42,43] is applied in three dimensions to model beam deflection and to gather the reaction forces in the anchoring structures. In a general framework, the problem can be described by the coupled equations:

$$\rho A \frac{\partial^2 w}{\partial t^2} - q(x,t) = \frac{\partial}{\partial x}\left[\kappa A G\left(\frac{\partial w}{\partial x} - \varphi\right)\right] \tag{2.2}$$

and

$$\rho I \frac{\partial^2 \varphi}{\partial t^2} = \frac{\partial}{\partial x}\left(EI\frac{\partial \varphi}{\partial x}\right) + \kappa A G\left(\frac{\partial w}{\partial x} - \varphi\right), \tag{2.3}$$

where the unknowns are the translational displacement of the beam $w(x,t)$ and the angular displacement $\varphi(x,t)$, $x$ is the coordinate of a point in the beam and $t$ the time parametrization. The parameters defining the model are the density of the beam material $\rho$, the cross-sectional area $A$, the elastic modulus $E$, the shear modulus $G$, the second moment of area $I$ and the Timoshenko shear coefficient $\kappa$. In the system of equation (2.2), the load is expressed as distributed load $q(x,t)$. Here, a solid cylinder of diameter $d$ and length $L$ is considered to represent the GCX unit. The values of $d$ and $L$ were defined based on the bending stiffness value (EI) [4].

In order to solve the Timoshenko beam theory, the finite-element method-based commercial software ABAQUS (Simulia, Dassault Systemes) is used. The model is discretized into 100 quadratic elements. The haemodynamic shear force is applied to 1/6 of the upper segment of the beams. This modelling choice is motivated by previous findings [25], demonstrating (1) that for a GAG of 150 nm, the 25 nm-long upper segment contributes to the 90% of the drag force and (2) that 96% of the bending moment arises from the forces acting on the 1/6 upper segment of the GAG. Further evidence supporting these observations demonstrates that the GCX layer acts as a dense porous medium to blood flow, and that is not able to penetrate the GCX layer in depth so that the velocity profile within the layer rapidly decreases from the luminal tip of GCX, vanishing close to the EC membrane surface [25]. To complete the definition of the problem, the binding of the GCX into the anchoring structures is modelled in terms of the Dirichlet boundary condition, with zero imposed displacements.

## 2.3. Quantitative analysis of GCX-mediated transmission of the haemodynamic shear forces

The fluid stimuli $F_{\text{shear}}$ (in terms of WSS applied to the surface of the upper segment of the GCX model), and the transduced mechanical forces $F_{\text{mem}}$, applied to the anchor point on the EC membrane, are calculated at each node on the luminal surface of the carotid bifurcation model. For a quantitative description of the transmission of the haemodynamic stimuli to the EC membrane, we propose here appropriate indicators. In detail, the time-averaged values of $F_{\text{shear}}$ and $F_{\text{mem}}$ magnitude along the cardiac cycle are considered

$$\text{TAF}_{\text{shear}} = \frac{1}{T}\int_0^T |F_{\text{shear}}\mathbf{s},t)|\,\mathrm{d}t \tag{2.4}$$

and

$$\text{TAF}_{\text{mem}} = \frac{1}{T}\int_0^T |F_{\text{mem}}\mathbf{s},t)|\,\mathrm{d}t \tag{2.5}$$

Mechanotransmission is evaluated in terms of force ratio ($F_{\text{ratio}}$), i.e. the cycle-average value of the ratio of the magnitude of the force transmission to the EC membrane $F_{\text{mem}}$ versus the haemodynamic stimulus $F_{\text{shear}}$:

$$F_{\text{ratio}} = \frac{1}{T}\int_0^T \frac{|F_{\text{mem}}\mathbf{s},t)|}{|F_{\text{shear}}\mathbf{s},t)|}. \tag{2.6}$$

Mapping the distribution of the $F_{\text{ratio}}$ at the luminal surface of the carotid bifurcation allows the evaluation of the amount of the haemodynamic shear stimulus transmitted to cell membrane.

To better clarify the GCX's role, and to clarify if and to which extent the GCX layer alters the dynamics of the haemodynamic stimuli transduced to the EC membrane, an analysis of the frequency

domain of the haemodynamic stimuli $F_{shear}$ and of the reaction forces $F_{mem}$ is proposed. To this end, the Fast Fourier Transform (FFT) [44,45] is applied to the time history of the magnitude of the forces under investigation. The time step of the computational haemodynamics simulation data, equal to 4 ms, dictated the sampling rate (250 Hz) of the FFT. Technically, the FFT library implemented in the software package MATLAB R2015b (The MathWorks Inc., Natick, MA, 2000) is adopted for analysis in the frequency domain. The history of the variable at the lumen is mapped from the local cartesian coordinate system to a two-dimensional reference coordinate system defined by two orthonormal vectors $\eta(\mathbf{s})$ and $\zeta(\mathbf{s})$, lying on the tangent surface of the lumen. $\zeta(\mathbf{s})$ is defined by the time averaged WSS vector $T^{-1} \int_0^T |\tau_w(\mathbf{s}, t)| \, \mathrm{d}t$ and $\eta(\mathbf{s})$ is uniquely defined by the cross-product of $\zeta(\mathbf{s})$ and the local unit vector normal to the surface $\mathbf{n}$. Mapping the WSS to this local reference system allows the representation of the quantities of interest in a frame of reference where they can be easily compared. The redistribution of spectral power in the frequency domain, a consequence of the GCX transduction of the fluid stimulus to the EC membrane, is evaluated by introducing two frequency-based operators, the Spectral Power Ratio (SPR),

$$\text{SPR} = \frac{\sum_{n=0}^{\infty} \left| \mathcal{F}\{|F_{mem}|\}(n2\pi f_0) \right|^2}{\sum_{n=0}^{\infty} \left| \mathcal{F}\{|F_{shear}|\}(n2\pi f_0) \right|^2}, \tag{2.7}$$

where $n$ is the harmonic number, and the Dominant Harmonic Ratio (DHR), defined as

$$\text{DHR}(\mathbf{s}) = \frac{\text{DH}|F_{mem}|}{\text{DH}|F_{shear}|}, \tag{2.8}$$

where $\text{DH}|F_k| = \max(\mathcal{F}\{|F_k|\})$, $k = \{mem, shear\}$, $\mathcal{F}\{|F_k|\}$ is the FFT of the time history of the magnitude of force $F_k$ and $f_0$ is the fundamental frequency. The SPR in equation (2.7), inspired by the recently proposed spectral power index [46], is the ratio of the spectral power of the transduced versus applied haemodynamic shear force. DH in equation (2.8) is the dominant harmonic indicator [47], and it is defined as the harmonic with the highest amplitude. The DHR is thus an indicator of the energy shift occurring in the frequency spectrum as a consequence of the transduction mechanism.

## 2.4. Evaluation of descriptors for shear-based forces and their transmission to the endothelial cell membrane

The haemodynamic hypothesis [48] underlying most of the current research on localizing factors of vascular disease relies on the attribution of the preferential development of atherosclerosis in arterial bifurcations to the atherogenic role played by low [49,50] and oscillatory WSS [51]. Based on the current haemodynamic theory and on simulated haemodynamics, two of the most widely considered WSS-based haemodynamic descriptors are computed to localize sites at the vessel wall resistant or susceptible to lesion onset and progression. Technically, the distributions at the luminal surface of the time averaged wall shear stress (TAWSS) and oscillatory shear index (OSI) are computed as follows:

$$\text{TAWSS}(\mathbf{s}) = \frac{1}{T} \int_0^T |\tau_w(\mathbf{s}, t)| \, \mathrm{d}t \tag{2.9}$$

and

$$\text{OSI}(\mathbf{s}) = \frac{1}{2} \frac{\left| \int_0^T \tau_w(\mathbf{s}, t) \, \mathrm{d}t \right|}{\int_0^T |\tau_w(\mathbf{s}, t)| \mathrm{d}t}, \tag{2.10}$$

where $\tau_w(\mathbf{s}, t)$ is the WSS vector at the generic location $\mathbf{s}$ at the vessel wall, and $T$ is the duration of cardiac cycle. Inspired by the definition of the OSI, we introduce here the Oscillatory Force Index (OFI):

$$\text{OFI}(\mathbf{s}) = 0.5 \left[ \frac{1 - \left| \int_0^T F_{mem}(\mathbf{s}, t) \, \mathrm{d}t \right|}{\int_0^T |F_{mem}(\mathbf{s}, t)| \, \mathrm{d}t} \right]. \tag{2.11}$$

OFI is a measure of the multidirectionality of the mechanical force $F_{mem}$ transduced to the EC membrane. In order to investigate in more depth, the impact that mechanotransduction has on the identification of atherosusceptible regions at the luminal surface, the co-localization of two generic

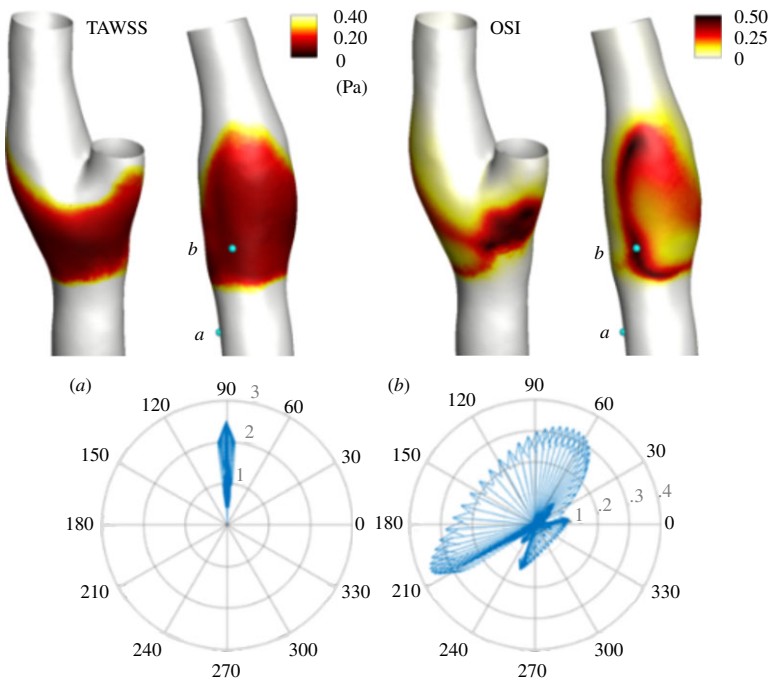

**Figure 1.** At the top, the OSI(**s**) and TAWSS(**s**) are plotted on the lumen of the carotid artery. White colour represents TAWSS > 0.40 Pa and OSI = 0. The most distinctive athero-protective (*a*) and athero-prone (*b*) locations are located. At the bottom, polar representations of $\tau(\mathbf{s}, t)$ for the locations (*a*) and (*b*) are represented.

descriptors *i* and *j* is performed by considering the overlap of the surface areas (SA) by applying the Similarity Index (SI), as already proposed elsewhere [52]:

$$\mathrm{SI}_{ij} = \frac{2 \cdot (\mathrm{SA}_i \cap \mathrm{SA}_j)}{\mathrm{SA}_i + \mathrm{SA}_j}, \tag{2.12}$$

where $\mathrm{SI}_{ij} = 0$ indicates that regions *i* and *j* have no spatial overlap, and $\mathrm{SI}_{ij} = 1$ that regions *i* and *j* are perfectly spatially overlapped. The considered SAs, the high OSI versus high OFI regions and the low TAF$_{\mathrm{shear}}$ versus low TAF$_{\mathrm{mem}}$ regions represent the regions exposed to values of the descriptors lower than the 20th percentile of the descriptor distribution for TAF$_{\mathrm{shear}}$ or TAF$_{\mathrm{mem}}$, and higher than the 80th percentile for OSI or OFI (being low and highly oscillating forces relevant in the context of atherosclerosis).

# 3. Results

## 3.1. Analysis of haemodynamic flow descriptors

We first solve the finite volume model as described in §2.1. Visualizations of WSS-based haemodynamic descriptors are presented in figure 1. It can be observed that high OSI values and low TAWSS values are both localized at the carotid bulb, in accordance with previous reports [34]. These regions are characterized by a low magnitude and rapid changes in the orientation of the WSS vector compared with the proximal region of the CCA, and the distal regions of ICA and ECA [34,36,51].

To further analyse how the flow behaves close to the lumen of these regions two locations are analysed; one with a high value of the OSI(s) and low TAWSS, associated with athero-prone regions, and the second with low value of the OSI(s) and high values of TAWSS, as an indicator of athero-protective regions (points A and B in figure 1). The time evolution of the WSS along time at these two locations is described in a polar plot in figure 1. Location A shows a maximum magnitude of the WSS along time of ≈2, 5 Pa and an almost unidirectional alignment of the WSS, but slightly oscillatory in magnitude, along the cardiac cycle. Location B shows large variation in the orientation of the WSS and the maximum magnitude of the WSS vector along the cardiac cycle did not reach 0.4 Pa. Overall, location A captures the distinctive features of athero-protective regions, i.e. high WSS magnitude and negligible variation in WSS direction, and location B captures characteristic low magnitudes and high direction oscillations of the WSS of athero-prone regions. In the light of

these results, specifically the WSS vector distribution and magnitude along time and space, how these highly varying fluid forces compare with a mechano-transduction through the GCX and ultimately with the forces exerted at the transmembrane anchors is investigated.

## 3.2. Identification of the mechanical forces acting in the glycocalyx-endothelial cell anchoring system

We apply the Timoshenko beam theory described in §2.2 to replicate (i) the results obtained by the continuum approach by Weinbaum and co-workers [25], considering a GCX of 200 nm length with a circular section of $d = 10$ nm that results in a value of bending stiffness EI = 490 kPa, and (ii) the MD model in [26,27], where a length of 50 nm and an EI value of 68 kPa were considered. Aiming at clarifying the impact that the Young modulus has on the deflection of the GCX, four different values of bending stiffness, i.e. EI = 68 KPa, reported in [26], 100 KPa, 490, reported in [25] and 700 KPa, are prescribed for the geometrical models described in [25]. The deflection values are 98.56, 68.01, 13.6 and 9.51 nm, respectively. The results for EI = 490 kPa are similar to the ones reported by the continuum approach but well above the ones reported by the MD simulations. Then, we analyse the geometrical model described in [26] through our continuum approach, keeping $d = 10$ nm. The deflection values of the beam is obtained when EI is set equal to 68, 100 and 700 kPa are 1.57, 1.086 and 0.15 nm, respectively. The results are out of the values reported by the MD simulations and the continuum model. Clearly, there are important discrepancies in the geometrical consideration and in the deflection results when both approaches are compared.

The differences in these two models are owing to the different structural components of the GCX. The GCX, as reviewed in [4], is mainly made of syndecans and HS. The HS are the structures exposed to the fluid and therefore the ones that transmit the forces to the transmembrane syndecans. The assumption of considering structures of 50 nm length and with a lower EI than the ones used in [25] is based on the fact that the here-considered HS-based model of GCX leads to more flexible structures that an averaged HS-syndecans network. In the MD simulation reported by Cruz-Chu *et al.* [26] and Jiang *et al.* [27], it is clear that HS are the ones that deform and transmit the forces, while the syndecans are actually just rotating due to the forces exerted by the HS. In conclusion, and given the discrepancies reported above, we consider the geometrical assumptions of the HS as in [27], which imposed a physiological range of velocities. We set the length of the beam to 50 nm and the value of EI to 100 KPa. The equivalent HS diameter $d$ in the continuum model is varied until the same deflections in the atomistic models for the different input loads in [26,27] are obtained. The identified value for the diameter of the beam is equal to 1 nm, one order of magnitude lower than the one indicated in [4]. In conclusion, we adopt a Timoshenko beam model of 50 nm in length, 1 nm in diameter and EI = 100 KPa, which is the combination of parameters that best reproduces the geometrical features and results reported in [26,27].

## 3.3. Characterization of shear and membrane forces in the frequency domain

The main features of $\tau_w(\mathbf{s}, t)$ in the frequency domain, which is relevant giving the oscillatory nature of the blood flow, are now analysed by means of the FFT. The two distinctive locations A and B defined in figure 1 are now analysed along the $\eta(\mathbf{s})$ and $\zeta(\mathbf{s})$ directions, defined in §2.3. The results of the FFT along both directions for the athero-prone and athero-protective locations are shown in figure 2. There is a clear transition in the range of $\log(10–12)$ Hz below which the high frequency–low amplitude WSS falls. At location A, all the values along the frequency domain along the $\eta(\mathbf{s})$ and $\zeta(\mathbf{s})$ directions are almost overlapping at $\approx 10^{-3}$ Pa. This result indicates a large variation in space of the orientation of $\tau(\mathbf{s}, t)$ along the cardiac cycle. At location B (athero-prone location), larger differences in the mean values of the WSS magnitude along the $\eta(\mathbf{s})$ and $\zeta(\mathbf{s})$ directions are found, indicating a low variation in the direction of $\tau(\mathbf{s}, t)$, as observed in figure 2. The mean value of the $\tau(\mathbf{s}, t)$ along its preferential direction is $\approx 10^{-2}$ Pa while its maximum, at low frequencies, is $\approx 0.5$ Pa.

To further investigate the mechano-transmission role of the GCX in non-uniform time-dependent flows, the FFT is again used to analyse the reaction forces at the anchor point of the GCX. The reaction forces at the anchoring structures of the GCX are computed as described in the previous section. Again, the locations A and B for the most distinctive athero-prone and athero-protective locations are analysed. The force generated at the anchoring proteins is presented in figure 2 in the frequency domain. The forces along the $\eta(\mathbf{s})$ and $\zeta(\mathbf{s})$ directions are again homogenized along the

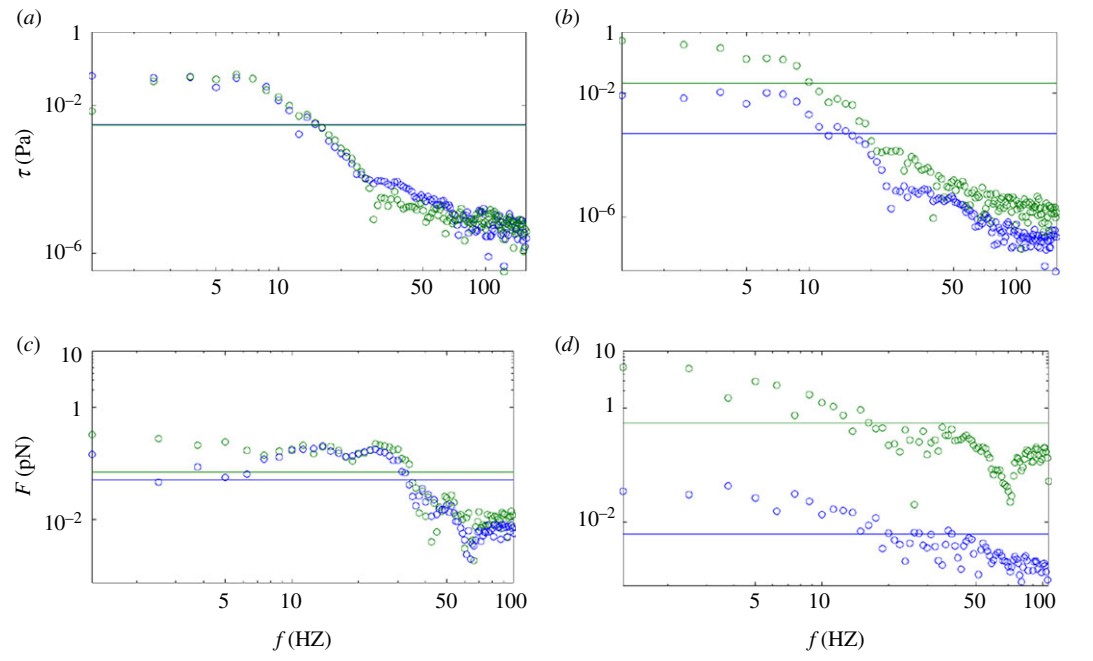

**Figure 2.** Panels (*a,b*) show the FFT of the WSS and (*c,d*) force vectors at the cell membrane in the $\eta(\mathbf{s})$ (green) and $\zeta(\mathbf{s})$ (blue) reference system. Panels (*a,c*) represent the FFT for the athero-prone and (*b,d*) athero-protective locations. Straight lines in each subfigure indicate the mean values of each dataset.

frequency domain and compared with the analysis of $\tau(\mathbf{s}, t)$. The data sample below the mean value is now under the $\approx \log(20-40)$ Hz range. At athero-prone locations, the reaction forces again show largest changes in their orientation and the maximum magnitude transmitted to the anchoring structures is $\approx 0.1$ pN. At the athero-protective location, the ratio between the mean magnitude of the reaction force along the $\eta(\mathbf{s})$ and $\zeta(\mathbf{s})$ directions is 100-fold. This result indicates that the GCX could act as a damping system, dissipating low magnitude–high frequency loads. The largest force is $\approx 5$ pN and the force magnitudes transmitted to the transmembrane anchors above 20 Hz are all in the range 1–10 pN.

## 3.4. Glycocalyx behaves as a dynamic damping system

A visualization of the time-averaged fluid stimuli $TAF_{shear}$ and the transduced force $TAF_{mem}$ at the EC membrane is presented in figure 3. The observed values of the fluid shear force $TAF_{shear}$ are higher than $TAF_{mem}$ force values at the inner wall of the internal (ICA) and external (ECA) carotid artery, at the apex of the bifurcation, and at the proximal CCA, with peak $TAF_{shear}$ and $TAF_{mem}$ values around 30 pN and 4 pN, respectively. Focusing on the bifurcation region, the distribution of the time-averaged value of the magnitude of the force transmitted to the transmembrane anchor $TAF_{mem}$ is, in general, more similar to the $TAF_{shear}$ distribution, in particular where the lowest force values can be observed (figure 3), i.e. at the carotid bifurcation outer wall corresponding to the enlargement of the ICA. In that region, on average along the cardiac cycle the magnitude of the transduced force at the membrane $F_{mem}$ predominates over the haemodynamic stimulus $F_{shear}$ ($F_{ratio} > 1$, in figure 3), whereas $F_{shear}$ magnitude predominates over $F_{mem}$ ($F_{ratio} < 1$, in figure 3) in the remaining regions where higher $F_{shear}$ magnitude values are observed. To complete the analysis, the co-localization of $TAF_{shear}$ and $TAF_{mem}$ is high (SI = 0.92), as well as the values of the 20th percentile of the distribution for $TAF_{shear}$ or $TAF_{mem}$ (0.199 pN versus 0.197 pN for $TAF_{shear}$ and $TAF_{mem}$, respectively).

To further analyse the GCX transmission of the fluid forces to the EC membrane, the distribution at the luminal surface of the two frequency-based descriptors SPR and DHR is presented in figure 4. Considering SPR, analogies between SPR and $F_{ratio}$ distributions can be observed: a higher spectral power is associated with the transduced force at the EC membrane (SPR > 1 in figure 4), corresponding to the aforementioned region characterized by $F_{ratio} > 1$ (figure 3). In this region, the GCX transmitted forces at the EC membrane are characterized by a lower dominant harmonic than the haemodynamic shear force (DHR < 1, figure 4), although associated with a higher spectral power. Contrarily, in the regions where $F_{ratio} < 1$ (figure 3), SPR values are lower than 1 while DHR values

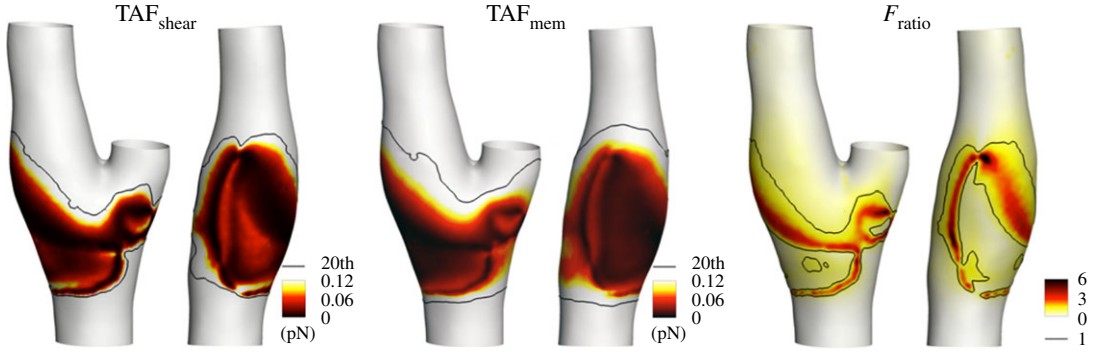

**Figure 3.** Distributions of the time-averaged shear force (TAF$_{shear}$), time-averaged membrane force (TAF$_{mem}$) and their ratio TAF$_{mem}$ over TAF$_{shear}$ ($F_{ratio}$) at the luminal surface (two different views). White colour represents TAF$_{shear}$ > 0.12 pN, TAF$_{mem}$ > 0.12 pN, $F_{ratio}$ = 0 respectively. The black isoline correspond to the value of the 20th percentile of the descriptor distribution for TAF$_{shear}$ or TAF$_{mem}$. For $F_{ratio}$, the isoline correspond to a value of 1 (i.e. TAF$_{mem}$ = TAF$_{shear}$).

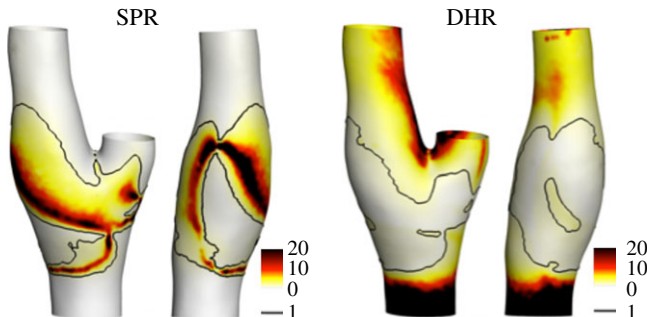

**Figure 4.** Distributions of the spectral power ratio (SPR), i.e. the ratio of the spectral power of the membrane force over the shear force (equation (2.7)), and dominant harmonic ratio (DHR), i.e. the ratio of the harmonic with highest amplitude for $F_{mem}$ and $F_{shear}$ (equation (2.8)) at the luminal surface (two different views). White colour represents SPR = 0 and DHR = 0. The black isoline corresponds to a value of 1.

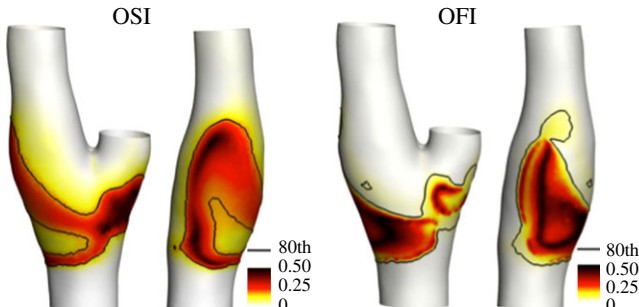

**Figure 5.** Distributions of the OSI (equation (2.10)) and oscillatory force index (OFI, equation (2.11)) at the luminal surface (two different views). The black isoline corresponds to the value of the 80th percentile of the descriptor distribution.

are larger than 1, indicating that in these regions a higher spectral power and a lower dominant harmonic is associated with the shear forces with respect to the GCX transmitted forces.

The impact of the GCX transduction in the force directional changes during the cardiac cycle is then evaluated by comparing contour maps of OSI and OFI at the luminal surface, presented in figure 5. The surface area exposed to OSI values larger than the 80th percentile value is located at the bifurcation region and covers the entire bulb at the proximal ICA, indicating that shear forces there undergo large directional changes during the cardiac cycle. Qualitatively, the surface area exposed to OFI values above its 80th percentile has a more limited extension in the ICA, as confirmed quantitatively by an SI value of 0.77, which indicates an only partial overlap of the surface area exposed to high OSI with the surface area exposed to high OFI. Moreover, lower values are found for the 80th percentile of the distribution of

OFI with respect to OSI (0.04 versus 0.16, for OFI and OSI, respectively). Therefore, these findings suggest that directional changes are attenuated as a consequence of the GCX mechanotransduction.

# 4. Discussion

The haemodynamic hypothesis underlying most of the current research on localizing factors of vascular disease relies on the attribution of the preferential development of atherosclerosis in arterial bifurcations to the atherogenic role played by low [49,50] and oscillatory WSS [51], described by OSI or the TAWSS descriptors. *In vitro* experimentation of the fluid features on EC is extensive but it has been mainly focused on laminar and sinusoidal oscillatory flows. Probably, the most complex set-up to reproduce complex pulsatile flows was developed by Blackman *et al.* [53]. However, complex variations in the orientation of the flow could not be imposed. As far as we know, no *in vitro* set-up has been proposed so far that can mimic the realistic and complex features of the blood flow *in vivo*. Therefore, we have used computational fluid dynamics to reproduce the realistic flow patterns in a three-dimensional reconstructed carotid artery.

However, how these forces are transmitted to mechano-sensing proteins of the GCX, which ultimately promote EC response to the fluid stimuli, have been widely overlooked. Experimental studies [1,9] and computational studies at atomistic [26] and continuum [25] levels have provided some insights into the mechanical behaviour of GCX. The GCX has been modelled by means of MD [26] as well as by continuum models of simplified periodic bush structures [25]. In this study, a multiscale approach is applied to analyse how near-wall fluid shear forces are transmitted to the transmembrane anchors of ECs, through the mechanical response of the GCX layer. Here, we analysed the GCX as a dynamic structure under the blood flow under the applied dynamic shear forces.

The presented analysis suggests that the presence of the GCX layer could alter the real forces acting on the anchoring elements at the membrane of ECs with respect to WSS stimuli, widely considered as a localizing factor of vascular disease. Notably, this study is intended to investigate the mechanical role of the GCX layer in transmitting haemodynamic shear forces to the EC membrane, assuming that the acting near-wall fluid forces are not disruptive of the GCX layer itself. Indeed, the study is intended to investigate the role of GCX mechanosensors at a pre-disease stage. This approach could also contribute to (at least partially) explaining the observed discrepancies in the location of regions with lesion prevalence and the luminal distribution of indicators of disturbed shear [54].

Among the main findings of the study, we report that: the cycle-averaged value of the magnitude of the force transmitted to the transmembrane anchor $F_{\mathrm{mem}}$ is, in general, similar to the distribution of the fluid shear force $F_{\mathrm{shear}}$, in particular in the proximal and distal region, where the lowest $F_{\mathrm{ratio}}$ values were reported. However, we found up to a six-fold increase of $F_{\mathrm{ratio}}$ in marginal regions of the carotid bifurcation in which the ratio between the $\mathrm{TAF}_{\mathrm{shear}}$ and $\mathrm{TAF}_{\mathrm{mem}}$ is six-fold, which demonstrate the different pattern of the fluid forces and the ones transmitted down to the cell membrane; secondly, we reported that the force dominant harmonic is lower in frequency at the EC membrane than at the fluid side and that the GCX affects the directionality of the transmitted forces; interestingly, we also found that the range of mechano-transduction force at the anchoring structures is $\approx 1\text{--}10\,\mathrm{pN}$. This result is remarkable as this range of force has been demonstrated in MD simulations [55,56] and *in vitro* experiments [57] to be effective for transition of conformational states of the vinculin-talin complex, and therefore of actin turnover and cell shape, under levels of force in the 2.5–20 pN range. Finally, in terms of localization factors, discrepancies between the OSI distribution and OFI distribution at the luminal surface can be observed, in particular in the bifurcation region (figure 1, top row). Altogether, these findings suggest that the GCX layer does not merely transmit near-wall forces to the EC membrane but modifies the sensed pattern of blood shear forces.

Still, some open questions remain and they will require further investigation. For example, in developed atherosclerosis the GCX is compromised. How the forces transmitted to the cytoskeleton will change under these circumstances is still not known. Here, we have assumed a uniform GCX layer. Some regions of the internal carotid sinuses of mice on high-cholesterol diet have shown a thinner GCX layer in comparison to the adjacent regions of the common carotid [58]. The idealized approach for modelling the GCX mechanical behaviour as a continuum, specifically for the HS protein, allows us to investigate the dynamic GCX response under shear flow all along the cardiac cycle. Our GCX model ameliorates the only available continuum model by Weinbaum and co-workers [25] by considering the inertial term. Moreover, the continuum approach, even if simplified, is a reasonable approach to explore fluid shear transduction to EC membrane all along the cardiac cycle. However, it also has the disadvantage of the need to select the

proper materials and adopt geometrical simplifications. More detailed strategies, such as the full atomistic approach recently proposed [27], cannot be applied to resolve the dynamic response of the GCX in the time scale of 1 s. Therefore, other types of coarse-grained models [28–30] will have to be developed to be integrated in large time and spatial scale models. Moreover, we calculated the forces exerted by blood on the glycocalyx using macro-scale data, jumping to the nanometre scale for modelling transmission of these forces to the membrane. The surface of the artery is not flat due to the thickness of the EC around its nucleus. We have not considered the intermediate scale of the artery's surface. The cell shape [59] and the interactions between constituents of the glycocalyx [27,60] may have an impact on both the magnitude and direction of force vectors experienced by the glycocalyx. A multi-scale approach to go across this wide range of length scales is out of the scope of this study. We focused on macroscopic quantities at the artery scale to cover a range of realistic haemodynamic forces, both in terms of magnitude and direction. Finally, how the mechanosensitive chain of EC induces specific cell shapes (round and tip-like) and F-actin organization is still not completely understood. Moreover, blood was assumed as a homogeneous Newtonian fluid with constant density (equal to 1060 kg m$^{-3}$) and viscosity of 3.5 cP. However, the GCX is in contact with the cell-free plasma as it is located at the luminal surface.

## 5. Conclusion

In conclusion, the approach proposed here could contribute to stimulate future studies on the mechanisms of transmission of local near-wall fluid forces sensed by the GCX to the EC membrane, thus bridging the gap of knowledge still existing. In particular, mapping near-wall forces distribution at the luminal surface versus the forces transmitted to the EC membrane could represent a powerful tool to link haemodynamics to the mechanobiology of the endothelium. The approach proposed here is based upon a simplistic description of the GX that could only partially capture the complexity of the transmission of fluid forces to the endothelium. However, the proposed approach has given us a snapshot of plausible mechanotransduction patterns throughout the whole carotid bifurcation with a moderate computational effort. More accurate GX modelling, e.g. based on full atomistic molecular dynamics simulations coupled with CFD, would be more appropriate for addressing a mechanistic description of the phenomenon, but the computational costs are prohibitively expensive. Despite the aforementioned limitations, the provocative intent and findings of the present *in silico* study warrant future investigations focusing on the actual forces transmitted to the transmembrane mechanotrasductors, which might outperform haemodynamic descriptors of disturbed shear as localizing factors of vascular disease.

Data accessibility. Data from the CFD simulation, the mechanical model of the glycocalyx, a MATLAB code of the flow and force descriptors and results on all descriptors presented in the document are freely from the Dryad. Data available from the Dryad Digital Repository: https://doi.org/10.5061/dryad.q83g0bv [61].
Author's contributions. P.S. designed the study, performed the computational mechanic simulations, interpreted the data and drafted the manuscript; D.G. designed the study, performed the computational fluid dynamic simulations, interpreted the data and helped draft the manuscript; U.M. designed and coordinated the study, interpreted the data and helped draft the manuscript. All authors gave final approval for publication.
Competing interests. We have no competing interests.
Funding. This work was supported by the Generalitat de Catalunya through the grant no. 2017-SGR-1278.

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
