## [Reviewer comments · Royal Society Open Science]

Review History

RSOS-190607.R0 (Original submission)

Review form: Reviewer 1

Is the manuscript scientifically sound in its present form?

Yes

Are the interpretations and conclusions justified by the results?

Yes

Is the language acceptable?

Yes

Is it clear how to access all supporting data?

No

Do you have any ethical concerns with this paper?

No

Have you any concerns about statistical analyses in this paper?

No

Recommendation?

Accept with minor revision (please list in comments)

Comments to the Author(s)

Code for the continuum model and force transduction calculations is not provided.

Review form: Reviewer 2

Is the manuscript scientifically sound in its present form?

Yes

Are the interpretations and conclusions justified by the results?

Yes

Is the language acceptable?

Yes

Is it clear how to access all supporting data?

Yes

Do you have any ethical concerns with this paper?

No

Have you any concerns about statistical analyses in this paper?

No

Recommendation?

Accept as is

Comments to the Author(s)

The Authors satisfactorily responded to the previous concerns.

Decision letter (RSOS-190607.R0)

10-May-2019

Dear Dr Saez

On behalf of the Editors, I am pleased to inform you that your Manuscript RSOS-190607 entitled "Mechanotransduction of hemodynamic forces by the endothelial glycocalyx in a full-scale arterial model" has been accepted for publication in Royal Society Open Science subject to minor

revision in accordance with the referee suggestions. Please find the referees' comments at the end of this email.

The reviewers and handling editors have recommended publication, but also suggest some minor revisions to your manuscript. Therefore, I invite you to respond to the comments and revise your manuscript.

- Ethics statement

- Data accessibility

If you wish to submit your supporting data or code to Dryad (<http://datadryad.org/>), or modify your current submission to dryad, please use the following link:
<http://datadryad.org/submit?journalID=RSOS&manu=RSOS-190607>

- Competing interests

- Authors' contributions

- Acknowledgements

- Funding statement

Because the schedule for publication is very tight, it is a condition of publication that you submit the revised version of your manuscript before 19-May-2019. Please note that the revision deadline will expire at 00.00am on this date. If you do not think you will be able to meet this date please let me know immediately.

Supplementary files will be published alongside the paper on the journal website and posted on the online figshare repository (<https://rs.figshare.com/>). The heading and legend provided for each supplementary file during the submission process will be used to create the figshare page,

so please ensure these are accurate and informative so that your files can be found in searches. Files on figshare will be made available approximately one week before the accompanying article so that the supplementary material can be attributed a unique DOI.

on behalf of Prof R. Kerry Rowe (Subject Editor)
openscience@royalsociety.org

Associate Editor Comments to Author:

Thank you for transferring this manuscript to RSOS and for providing such effective responses to the earlier reviewers. Please note that acceptance is contingent on you providing the code etc. requested by the referees -- if you do not provide this information, we cannot proceed.

Reviewer comments to Author:
Reviewer: 1

Comments to the Author(s)
Code for the continuum model and force transduction calculations is not provided.

Reviewer: 2

Comments to the Author(s)
The Authors satisfactorily responded to the previous concerns.

Author's Response to Decision Letter for (RSOS-190607.R0)

See Appendix A.

Decision letter (RSOS-190607.R1)

14-May-2019

Dear Dr Saez,

I am pleased to inform you that your manuscript entitled "Mechanotransduction of hemodynamic forces by the endothelial glycocalyx in a full-scale arterial model" is now accepted for publication in Royal Society Open Science.

on behalf of R. Kerry Rowe (Subject Editor)
openscience@royalsociety.org

Follow Royal Society Publishing on Twitter: [@RSocPublishing](https://twitter.com/RSocPublishing)

Appendix A

Referee: 1

Comments to the Author

This work uses computational modeling to estimate the effects of the glycocalyx on the forces exerted by blood flow on the endothelial cells lining a bifurcation in the carotid artery. The manuscript describes a multiscale approach that allows consideration of flow over the time scale of the cardiac cycle while examining forces spatially at the nanometer level. Estimates of forces at the cell membrane are provided, and interesting differences are reported between the patterns of forces exerted by blood on the glycocalyx and the patterns transmitted to the cell membrane by the glycocalyx. Taken at face value, these calculations offer novel insight into the cell/molecular mechanisms that might be involved in endothelial responses to blood flow.

First of all, we would like to thank you for the positive comments and criticism to our work, which will increase its quality.

One major concern is that the analysis calculates the forces exerted by blood on the glycocalyx using macro-scale data and jumps directly to the nanometer scale for modeling transmission to the membrane without consideration of the intermediate, micrometer-scale structure of the vessel's surface. The surface is not flat; it has considerable bulges into the lumen due to the greater thickness of each endothelial cell around its nucleus. This is likely to have considerable impact on both the magnitude and direction of force vectors experienced by the glycocalyx, which is of particular concern when the force transmission to the cell membrane is being modeled on the nanometer scale. This issue should be addressed.

Thank you for the comment. We agree with the reviewer. We calculated the forces exerted by blood on the glycocalyx using macroscale data, jumping into the nanometer scale for modeling transmission to the membrane. The surface of the artery is not flat due to the thickness of the endothelial cell around its nucleus. However, we have not considered the intermediate scale of the artery's surface and we based our results in the fluid forces exerted on the most exposed, upper side of the EC. The cell shape and the interactions between constituents of the glycocalyx may have an impact on both the magnitude and direction of force vectors experienced by the glycocalyx. A multi-scale approach to go across this wide range of length scales is out of the scope of this study. Note that we are solving the problem, even if using an idealized model of mechanotransduction, in every node of the computational mesh of ~37000 nodes. Here our intent was provocative: can we demonstrate, even if with an idealized/simplified approach, that the paradigm of computational hemodynamics, widely applied to identify regions of disturbed shear at the luminal surface of arterial segments, is adequate to describe the hemodynamic risk of vascular disease? Could we have a different pattern of atherosclerotic risk at the arterial wall, considering that near wall hemodynamic forces are not transmitted as they are to the endothelial cells? The simplified approach adopted in this study allowed us to address this questions in a large range of conditions, highlighting the differences in the response of the GCX to the wide variety of hemodynamic stimuli in a realistic geometry.

To compute the shape effect of the cell, we should now develop a multiscale method in which every node would correspond with the position of a cell, and do a new simulation in this new domain, which would increase considerably the computational time. Here, we focused on macroscopic quantities at the artery scale. We consider the most exposed section of the cell, on top of the dome, which are the section more exposed to the fluid forces. We have also included references of numerical studies that have looked at this aspect in one single cell (reference 14 and 30 in the revised manuscript). We believe that these references will complement our contribution with an intermediate scale perspective. We have added a discussion about this point in the discussion of the manuscript, highlighted in blue in the revised manuscript

Additional points needing clarification:

In Figures 1, 3 and 4, do the white regions on the arteries represent no data being displayed or do they indicate that the values are above or below the color scale? White regions are seen on both sides of some of the isolines in figures 3 and 4, suggesting that white represents a calculated value, but the uniform white around the bifurcation apex in most of the figures suggests that data is simply not being displayed. The boundaries of the data representation in these images should be clarified, perhaps with a different color or pattern if there are regions that are included to show the geometry but do not have force values.

Thank you for pointing this out. We apologize for not being more precise in the first place. White regions have out of scale values, as white is one limit of the color scale. We have modified the figure captions to describe what white regions represent in each figure.

The sentence on p. 6, lines 8-10, describing the coordinate system appears to give 2 conflicting definitions of the vector, ζ . Should the second definition be referring to the vector η ?

Yes, this is a typo. We have corrected it in the current version of the manuscript.

The paper's title is grammatically confusing. Is mechanotransduction really "of the endothelial glycocalyx" and "to the hemodynamic shear forces"? Is the intended meaning "mechanotransduction of hemodynamic forces by endothelial glycocalyx"?

Thank you for the suggestion. We have rephrased the title as suggested.

Referee: 2

Comments to the Author

This submitted manuscript seeks to understand better the magnitude that blood flow induced shear forces are transmitted to the endothelial cell transmembrane structures via the glycocalyx. Utilizing predicted temporal hemodynamic profiles derived from image-based modeling of a human

carotid bifurcation, the Authors utilize Timoshenko beam theory to quantify the transmission of shearing forces from the tip to the base of generalized cylindrical beams (continuum model), which are representative of the heparan sulfate structures in the glycocalyx. Evaluated metrics include force transduced to the cellular membrane, ratio of applied (fluid shearing) to transduced force, spectral power ratio (frequency domain quantity), oscillatory force index (OFI), and the similarity index between varying magnitudes of OFI and oscillatory shear index (OSI) regions. Results demonstrate a large difference between computed near-wall hemodynamic metrics and value transmitted to the cellular surface. Furthermore, differences in the degree of temporal oscillation are observed between near-wall shearing stresses and transmitted forces. Thus, the Authors conclude that the glycocalyx not only dampens the magnitude of forces that are actually transmitted to the cellular surface, but may alter the (temporal) shear patterns transmitted.

We thank the reviewer for the constructive criticism and for the time spent in improving our work.

The Authors present an interesting study with intriguing findings; however, there are concerns that the continuum model is oversimplified in both its geometry and interaction with surrounding structures. Specific comments include:

1. The computational geometry seems extremely simplified, as compared to the native structure of the glycocalyx. The glycocalyx is a complex structure, with a heterogeneous distribution of membrane bound proteoglycans and glycoproteins. This manuscript only includes a single protein, heparan sulfate, and utilized a generalized geometry for its structure that differs tremendously from the true form. While molecular dynamic simulations lack the ability to examine the time over a cardiac cycle, these simulations offer greater physical representation of these microstructures. For example, Jiang et al. (ref. 29) observed significant interactions between constituents of the glycocalyx that impact resulting shear stress profiles. The inherent assumptions in this manuscript limit the enthusiasm and confidence in the presented results.

We agree with the reviewer's point of view. The approach proposed here is based upon a simplistic description of the GCX that could only partially capture the complexity of the transmission of fluid forces to the endothelium.

However, the proposed approach has allowed having a snapshot of plausible mechanotransduction patterns throughout the whole carotid bifurcation with a moderate computational effort, by exploring a wide range of hemodynamic WSS profiles. More accurate GCX modeling, e.g., based on full atomistic molecular dynamics simulations coupled with CFD, would be more appropriate for addressing a mechanistic description of the phenomenon at the single molecule scale. However, it is not possible, as the reviewer points out, to get information along the entire cardiac cycle due to time-scale limitations. Even coarse grained modelling is not a viable solution, as computing all possible hemodynamic fluid patterns, as we have done here for the simplified model, would be extremely computationally and time-expensive.

Despite the aforementioned limitations, the provocative intent and findings of the present *in silico* study warrant future investigations focusing on the actual forces transmitted to the transmembrane mechanotransducers, which might outperform hemodynamic descriptors of disturbed shear as localizing factors of vascular disease. We believe that these findings are meaningful and could open new investigation in this direction. We have added these arguments in the Discussion section of the revised version of the manuscript.

2. The results presented in Section 3.2 further strengthen the argument that the continuum assumption is not ideal for this problem. When considering similar beam dimensions and stiffness values, variations in beam deflections are observed. What is the reasoning behind this difference, and how does it affect the subsequent? Understanding this discrepancy is key for confidence in follow-up studies.

We use $EI = 68 \text{ KPa}$, 100 KPa , 490 kPa and 700 KPa . Then we also varied the diameter of the structure. In all of them we got different deflection values. As a consequence, the dimension parameters are different and thus different deflections are, of course, obtained.

The goal of that part of the study was to consider some length dimension that are known from experimental results (HS of 50 nm in length) and predict the other parameters, resulting in a structure of 1 nm in diameter and $EI = 100 \text{ KPa}$. This combination of parameter that better reproduced the geometrical features and deflection results reported in [12, 30].